# Transcriptomic Analysis in Human 3D Skin Model Injected with Resorbable Hyaluronic Acid Fillers Reveals Foreign Body Response

**DOI:** 10.3390/ijms232113046

**Published:** 2022-10-27

**Authors:** Danyel G. J. Jennen, Marcel van Herwijnen, Marlon Jetten, Rob J. Vandebriel, Peter Keizers, Robert E. Geertsma, Wim H. de Jong, Jos C. S. Kleinjans

**Affiliations:** 1Department of Toxicogenomics, GROW School for Oncology and Reproduction, Maastricht University, 6200 Maastricht, The Netherlands; 2Centre for Health Protection, National Institute for Public Health and the Environment (RIVM), 3720 Bilthoven, The Netherlands

**Keywords:** 3D skin model, hyaluronic acid, dermal fillers, transcriptomics, inflammation

## Abstract

Usage of injectable dermal fillers applied for aesthetic purposes has extensively increased over the years. As such, the number of related adverse reactions has increased, including patients showing severe complications such as product migration, topical swelling and inflammatory reactions of the skin. In order to understand the underlying molecular events of these adverse reactions we performed a genome-wide gene expression study on the multi-cell type human Phenion^®^ Full-Thickness Skin Model exposed to five experimental hyaluronic acid (HA) preparations with increasing cross-linking degree, four commercial fillers from Perfectha^®^, and non-resorbable filler Bio-Alcamid^®^. In addition, we evaluated whether cross-linking degree or particle size of the HA-based fillers could be associated with the occurrence of adverse effects. In all cases, exposure to different HA fillers resulted in a clearly elevated gene expression of cytokines and chemokines related to acute inflammation as part of the foreign body response. Furthermore, for one experimental filler genes of OXPHOS complexes I-V were significantly down-regulated (adjusted *p*-value < 0.05), resulting in mitochondrial dysfunction which can be linked to over-expression of pro-inflammatory cytokines TNFα and IL-1β and chemokine CCL2. Our hypothesis that cross-linking degree or particle size of the HA-based fillers is related to the biological responses induced by these fillers could only partially be confirmed for particle size. In conclusion, our innovative approach resulted in gene expression changes from a human 3D skin model exposed to dermal fillers that mechanistically substantiate aforementioned adverse reactions, and thereby adds to the weight of evidence that these fillers may induce inflammatory and fibrotic responses.

## 1. Introduction

Over recent years, the number of non-surgical aesthetic procedures has increased tremendously. Between 2015 and 2019, the International Society of Aesthetic Plastic Surgery [1] reported a worldwide increase of the number of procedures using hyaluronic acid (HA) fillers of 50.6%. In the Netherlands the number of treatments with injectable soft tissue fillers (dermal fillers) has increased by approximately 12%, with over 160,000 treatments on a population of 11.8 million persons 18–70 years of age eligible for treatment over a three-year period from 2016 to 2019 [2].

With the increase of procedures, the range of different dermal filler products on the market also increased. Currently, approximately 160 dermal filler products, produced by more than 50 different manufacturers, are available worldwide [3]. The most popular types of dermal fillers are HA-based because of their ease of application, and satisfactory aesthetic results [4]. In addition, they were found not to be toxic in a wide range of acute animal toxicity studies [5,6], nor did they cause in vitro cytotoxicity in human or mouse cell lines [4,6,7]. However, complications do occur after injections with HA, as well as with other dermal fillers [8] The calculated risk for serious HA filler complications is 0.03%, of which vascular adverse events are the most serious ones, with an occurrence of 0.014% [2]. The adverse reactions of the HA-based fillers are mainly swelling, erythema, and nodules [9,10,11]. Incidentally, even late-occurring adverse effects (e.g., delayed onset nodules, granulomatous foreign body reaction (FBR), severe edema) have been reported [12,13,14]. Given the large variability in the HA-based fillers concerning HA sources (primarily rooster comb or bacterial), cross-linkage (agent and degree), HA concentration, hardness, cohesivity, consistency, and longevity of the resulting correction [15], it is unclear what causes the adverse effects, i.e., the relation between product characteristics and adverse effects is not well understood. Consequently, more knowledge is needed on the type of potential complications and the molecular mechanisms underlying the induction of these adverse responses. This would be best studied in skin samples (biopsies) from patients. However, procedures to obtain biopsies are invasive and can lead to scarring or other severe complications [16] and, thus, patients are reluctant to give consent for biopsies to be taken, especially coming from the facial area (Decates, personal communication). Moreover, there is no need for an operational procedure at which biopsies could be taken, as treatment of the patients with hyaluronidase, an enzyme that degrades HA-based fillers, results in a decrease of the swelling without leaving (severe) traces of the procedure [17,18,19].

An alternative to patient material would be the use of animals, which are currently required anyway for the evaluation of the HA-based fillers as medical devices. However, animal to human translation remains difficult as animals differ from humans in their genetic makeup, in their environmental exposure, and in their phenotype [20,21,22,23]. Therefore, the best alternative to study adverse effects of soft tissue filler injections on the molecular level are in vitro cell models from human origin. In the study of De Jong et al. [7], we used human THP-1-derived macrophages to investigate the molecular changes after exposure to a range of resorbable HA-based fillers, i.e., a series of five experimental HA preparations synthesized with an increasing degree of cross-linking and four commercial fillers of the Perfectha^®^ product line. Furthermore, the macrophages were exposed to the non-resorbable filler Bio-Alcamid^®^. The results of that study showed an initial immune response of the macrophages exposed to the dermal fillers and the exposure to Bio-Alcamid^®^ also on the gene expression level. In addition, on the gene expression level, cell cycle arrest was observed for the exposure to the various HA fillers, which may indicate an alarm response to prevent the propagation of dysfunctional cells. In that study, we did not find a clear relation between the different characteristics of the examined HA-based fillers and the molecular responses induced by them.

The results of the THP-1-derived macrophages exposed to the dermal fillers only reflect an early reaction of the so-called FBR that eventually may lead to the formation of foreign body giant cells (FBGC) and fibrotic encapsulation [24,25]. To investigate a wider specter of adverse processes from a multitude of interactions of the various cell types in skin tissue, a more sophisticated in vitro system is required. In the current study, we aimed to unravel the molecular events caused by exposure to the dermal fillers in a multi-cell type human 3D skin model and to link these events to the characteristics of the dermal fillers. Therefore, we investigated the gene expression changes in the human Phenion^®^ Full-Thickness Skin Model after injection of the five experimental HA preparations with an increasing amount of cross-linking, the four commercial fillers from the Perfectha^®^ product line differing in particulate size, and the non-resorbable filler Bio-Alcamid^®^. The Phenion^®^ model is a 3D skin model reconstructed from primary human foreskin keratinocytes and fibroblasts, consisting of an epidermis, a basement membrane, and a dermis, which shows histological and physiological properties of native human skin [26]. We hypothesized that the exposure of the human Phenion^®^ Full-Thickness Skin Model to dermal fillers with a higher degree of cross-linking or with a larger particle size would cause a more profound adverse effect, which would be evident in the gene expression response.

## 2. Results

### 2.1. Identification of DEGs in the Phenion^®^ Full-Thickness Skin Model

The numbers of differentially expressed genes (DEGs) based on an absolute FC > 1.5 and an adjusted *p*-value < 0.05 for the in vitro exposed human full thickness skin model are shown in Table 1 and Table 2. Furthermore, for those comparisons showing a low number of DEGs for an adjusted *p*-value < 0.05, a less stringent cut-off was used, i.e., an adjusted *p*-value < 0.2. The full output of the gene expression analysis is available as Appendix A.

### 2.2. Pathway Analysis

An ORA in PathVisio was performed for the selected DEGs of the in vitro experiments. In Table 3, Table 4, Table 5 and Table 6, the top significant pathways are shown for the different gene sets. A full overview of the pathway analysis results is available as Appendix A. Cut-off values, i.e., adjusted *p*-values, for DEG selection using moderated *t*-test are indicated. Moreover, for Bio-Alcamid^®^ and RIVM 2 preparation, an adjusted *p*-value < 0.05 was applied (Table 3 and Table 4, respectively) and for RIVM 1, 3, 5 preparations and Perfectha^®^ SubSkin, an adjusted *p*-value < 0.2 was applied (Table 5 and Table 6, respectively). No pathway analysis was conducted for the RIVM 4 preparation and Perfectha^®^ FineLines, Derm, and Deep, as the number of DEGs was too low, even when applying the less stringent adjusted *p*-value < 0.2. Overall, in the pathway analysis, the gene expression responses of the Phenion^®^ Full-Thickness Skin Model exposed to any of the remaining fillers resulted in immune-related pathways indicative of cytokine activity and inflammation. The gene expression changes for Bio-Alcamid^®^, the RIVM preparations, and the Perfectha^®^ fillers are visualized on the “Cytokines and inflammatory response” pathway and on the “Overview of proinflammatory and profibrotic mediators” (Figure 1 and Figure 2, respectively). For most fillers, up-regulation of the various cytokines and chemokines is evident, with Bio-Alcamid^®^ and RIVM 2 preparation showing the highest expression. Furthermore, a higher expression is seen for Perfectha^®^ Deep and SubSkin compared to Perfectha^®^ FineLines and Derm, which could be an indication that particle size may have an influence on the gene expression changes.

In addition to, the immune-related pathways, Phenion^®^ exposed to the RIVM 2 preparation also showed gene expression changes related to cellular respiration. In Figure 3, these gene expression changes for the RIVM 2 preparation as well as for Bio-Alcamid^®^ and the other RIVM preparations are visualized on the “Electron transport chain: OXPHOS system in mitochondria” pathway, clearly indicating that energy production through the mitochondria is inhibited as most genes of the oxidative phosphorylation (OXPHOS) complexes I–V are down-regulated. In addition, the “Glycolysis and gluconeogenesis” pathway was also significantly changed for the RIVM 2 preparation exposure, i.e., several genes involved in the glycolysis were up-regulated, whereas those involved in the gluconeogenesis were down-regulated.

## 3. Discussion

This study investigated the effects on global gene expression induced by resorbable HA-based fillers as well as the non-resorbable filler Bio-Alcamid^®^ in the human Phenion^®^ Full-Thickness Skin model, a 3D skin model with a similar morphology and a comparable expression of several differentiation markers as native human skin. In this model, the alterations in various immune related pathways were evident. A clear inflammatory response, showing up-regulation of cytokines and chemokines (Figure 1 and Figure 2), was observed for the Phenion^®^ model after exposure to all RIVM preparations and Perfectha^®^ fillers, with RIVM 2 and Perfectha^®^ SubSkin, respectively, showing the highest gene expression changes. In addition, the exposure to Bio-Alcamid^®^ also exerted an increased inflammatory response. This response can be considered to reflect an acute inflammatory response, as observed in patients injected with HA fillers [27]. It also depicts the start of the FBR; within hours after injection, M1 macrophage differentiation takes place and proinflammatory actors, such as TNFα (tumor necrosis factor α), and interleukins IL-1β, IL-6 and IL-8 (also known as CXCL8) are released [25]. This event was confirmed by our gene expression analyses in which TNFα was significantly up-regulated (adjusted *p*-value < 0.05) for Bio-Alcamid^®^, and IL-1β, IL-6, and IL-8 were significantly up-regulated (adjusted *p*-value < 0.05) for Bio-Alcamid^®^ and RIVM 2 preparation. The other fillers showed an elevated expression for TNFα, IL-1β, IL-6, and IL-8 genes. In addition, several chemokines from the CC and CXC subfamilies were also elevated for all fillers and significantly up-regulated (adjusted *p*-value < 0.05) for Bio-Alcamid^®^ and/or RIVM 2 preparation (Figure 2). CCL2, significantly up-regulated for both fillers, is a profibrotic chemokine, which participates in macrophage fusion and foreign body giant cell formation [28], and is thus an indicator of the next step in the FBR [25]. In this step, TGF-β can also be released, thereby initiating fibrotic encapsulation, the final step of the FBR [25]. TGF-β was significantly up-regulated (adjusted *p*-value < 0.05) for RIVM 2 preparation.

For the exposure to the Perfectha^®^ fillers gene expression changes were lower, but nonetheless many cytokines and chemokines were up-regulated. In general, a higher gene expression was evident for Perfectha^®^ Deep and SubSkin compared to Perfectha^®^ FineLines and Derm. Moreover, a few genes, like IL-6, showed an increase in expression when the particle size also increased (Figure 1). This trend was also seen for the anti-inflammatory genes IL-10, IL-19, IL-20, and IL-24 (Figure 2). IL-10 is released by M2 macrophages in the differentiation step of the FBR, causing decreased inflammatory activity [25]. Whether the increased particle size leads to more M2 macrophages that are also involved in the formation of FBGC and, thus, more IL-10 expression, is unknown and remains to be determined. In the Phenion^®^ Full-Thickness Skin model no macrophages were included and thus the origin of the changed IL-10 expression is unclear. IL-19, IL-20, and IL-24 belong to the IL-20 subfamily of cytokines, which on the one hand, act on epithelial tissues, such as keratinocytes, to enhance epithelial innate immunity and amplify inflammatory responses and, on the other hand, they have unique roles in driving tissue protection and regeneration [29]. This potential effect of the IL-20 subfamily of cytokines was only observed for the exposure to the Perfectha^®^ fillers as IL-10, IL-20, and IL-24 were not expressed for Bio-Alcamid^®^ nor the RIVM preparations.

Alongside the immune related pathways, the Phenion^®^ Full-Thickness Skin model exposed to the RIVM 2 preparation also showed significantly down-regulated genes (adjusted *p*-value < 0.05) of the OXPHOS complexes I-V resulting in mitochondrial dysfunction (Figure 3). On the other hand, the exposure to RIVM 2 resulted in increased expression of the genes involved in glycolysis, thus still providing a source of energy. In other words, these results indicate that a shift from an OXPHOS-dependent ATP production to a glycolytic ATP production took place, a process also seen in M1 macrophage differentiation [30].

Mitochondria play an important role in skin homeostasis, particularly in epidermal differentiation. Moreover, renewal of the epidermis depends on keratinocyte differentiation, for which mitochondrial respiration and ROS production are essential [31]. As such, mitochondrial dysfunction has been shown to increase proinflammatory cytokine production [32] and chemokine CCL2 overexpression has also been linked to altered mitochondrial dynamics [33]. As indicated before, increased expression of the proinflammatory cytokines TNFα and IL-β as well as of chemokine CCL2, was also observed in our study and can be directly linked to the mitochondrial dysfunction observed for the RIVM 2 preparation injection into the Phenion^®^ model. Down-regulation of the OXPHOS genes in the Electron transport chain: OXPHOS system in mitochondria pathway was also observed for the other RIVM preparations and Bio-Alcamid^®^, but for the exposure to the Perfectha^®^ fillers the gene expression changes were less profound. Therefore, it is unclear to what extent mitochondrial dysfunction occurs for these fillers.

Although our results clearly show that exposure of the human Phenion^®^ Full-Thickness Skin model to the different dermal fillers leads to an acute inflammatory response, we expected that this response could be related to the different characteristics of the dermal fillers. To some extent, we did show that size matters as higher gene expression levels of the cytokines and chemokines were observed for the exposures to the Perfectha^®^ fillers Deep and SubSkin with larger sized particles compared to the Perfectha^®^ fillers FineLines and Derm with smaller sized particles. For the degree of cross-linking, the relationship with the gene expression changes is not clear as the gene expression levels for the exposure to the different RIVM preparations were not in concordance with the modification and cross-linking grades of these fillers, as was expected based on the study by Keizers et al. [34]. Moreover, for the RIVM 2 preparation, the gene expression changes were the largest. It should be noted that due to high viscosity the RIVM 3–5 preparations were difficult to inject even after diluting them. Whether this high viscosity and difficulty to inject the fillers resulted in a poor distribution of the filler in the skin tissue is unknown, but it could be a plausible reason for the low number of DEGs for the RIVM 3–5 preparations.

In conclusion, our hypothesis that the degree of cross-linking in the experimental HA preparations or the particle size in the commercial HA fillers is related to the biological responses induced by the HA-based fillers could only partially be confirmed for the particle size. Nonetheless, gene expression changes for the exposures of the human Phenion^®^ Full-Thickness Skin model to different HA-based fillers and Bio-Alcamid^®^ revealed acute inflammatory effects as an initial step of the FBR. Furthermore, indications of the progression of this FBR as well as a potential tissue regeneration after an insult, were also seen. However, it should be noted that in our experiments the Phenion^®^ model was only exposed for 24 h, whereas the macrophage fusion, foreign body giant cell formation, and fibrotic encapsulation usually takes place days or weeks after injection [25]. Therefore, our findings can be considered the tip of the iceberg. Additional experiments to unravel the molecular mechanisms occurring in the 3D skin model after dermal filler injection using longer exposure times and sampling at multiple time points are advised. In addition, our current setup of the experiments with the human Phenion^®^ Full-Thickness Skin model can be further improved by including macrophages in the medium [35] and/or by introducing vascularization to the system, by which a more physiological transport of nutrients can be established equivalent to real human skin tissue [36]. The current results combined with those from additional experiments present an opportunity for the development of gene expression profiles that can be predictive for HA-based filler material responses in patients.

## 4. Materials and Methods

### 4.1. Dermal Fillers

#### 4.1.1. Perfectha^®^

The commercial Perfectha^®^ product line (Sinclair Pharmaceuticals, Paris, France) comprises five HA dermal fillers, of which four were used. These fillers are distinguishable by their particle size, but not according to modification and cross-linking grade. For Perfectha Deep (8000 particles per mL), a uniform particle size was reported as 500 µm [6]. The Pefectha^®^ fillers contain 96% cross-linked HA and 4% HA that is not cross-linked HA according to the device description of the manufacturer (Laboratoire ObvieLine, Sinclair, Dardilly, France). A more detailed product description of these fillers is shown in Table 7. Each filler was provided in a concentration of 20 mg/mL.

#### 4.1.2. RIVM Preparations

Five HA preparations with differences in cross-linking degree were synthesized by the National Institute for Public Health and the Environment, Bilthoven, The Netherlands (Rijksinstituut voor Volksgezondheid en Milieu, RIVM) and designated RIVM 1–RIVM 5. The synthesis and analysis of these experimental HA fillers were described previously in Keizers et al. [34]. HA from rooster comb was treated with 1,4-butanediol diglycidyl ether (BDDE) according to Guarise et al. [37]. The details of this series are presented in Table 8.

#### 4.1.3. Bio-Alcamid^®^

Bio-Alcamid^®^, a non-resorbable dermal filler consisting of a three percent polyalkylimide suspension in water, was provided by Dr. Tom Decates (Erasmus MC, Rotterdam, The Netherlands) and was used as a filler with well-known long-term adverse effects.

### 4.2. Genome-Wide Evaluation of Effects on Gene Expression

#### 4.2.1. The Human Phenion^®^ Full-Thickness Skin Model

The Phenion^®^ Full-Thickness Skin Model (Henkel AG & Co, Düsseldorf, Germany) is a 3D model reconstructed from primary human foreskin keratinocytes and fibroblasts originating from one donor.

The Phenion^®^ model was handled and cultured under sterile conditions, applicable to standard cell culture conditions. Upon arrival the tissues were contained in a 24-well plate containing a semi-solid transport medium and a plastic insert to prevent damage during transport. After arrival, the tissues were immediately removed from their transport plate according to the manufacturer’s instruction. In short, culture dishes were prepared in a manner that allowed the culture of each tissue separately. The Air-Liquid interface (ALI)-cell culture medium, culture dishes, culture supports, and filter papers were all provided with the tissues by the manufacturer. A 24 or 48 h recovery period at standard cell culture conditions (37 °C in a humidified atmosphere containing 5% CO_2_) was implemented after transfer, after which all culture medium was removed and replaced by fresh ALI-cell culture medium (37 °C). Every tissue used in these experiments was of identical age at this point and could be cultured for at least another 10 days.

The tissues were exposed to the fillers by injecting ~0.3 mL directly under the top layer in the epidermal layer using a cross-hatching technique (Bray et al. 2010), as shown in Figure 4. The cross-hatching pattern ensures an equal distribution of the fillers in the whole tissue model. The commercial fillers Perfectha^®^ and Bio-Alcamid^®^ were injected as provided. The RIVM preparations (#1 to #5) were diluted with PBS 1:1 in order to be able to inject them. Unexposed tissue was used as control. Exposures lasted for 24 h at 37 °C and were performed in triplicate. Each set of exposures was performed on a different date using a fresh batch of tissue models, all constructed from the same donor. Thus, for each exposure three biological replicates were made.

#### 4.2.2. RNA Isolation

After exposure, the skin tissues were collected and cut into 4 quarters. Two opposite pieces were used for RNA isolation, whereas the other two pieces were stored at −80 °C. The skin pieces for RNA isolation were submerged in 1 mL pre-chilled (4 °C) QIAzol (Qiagen, Venlo, The Netherlands). A T10 basic ULTRA-TURRAX^®^ (IKA, Staufen, Germany) was used to homogenate the tissue in about 30 s, after which the solution was kept on ice until further processing (no longer than 15 min). Total RNA was isolated from the homogenates using the miRNeasy isolation kit (Qiagen, Venlo, The Netherlands) as instructed by the manufacturer. After isolation, RNA concentrations were measured on a NanoDrop spectrophotometer and the integrity was determined with a Bioanalyzer 2100 (Agilent Technologies, Amstelveen, The Netherlands). Only samples of good quality (clear 18S and 28S peaks and the RNA integrity number (RIN) > 6) were used for hybridization. An average RIN > 9 was measured. Extracted RNA was stored at −80 °C until it was used as template for cDNA synthesis.

#### 4.2.3. Microarray Preparation

From each sample, 0.2 μg of total RNA was used to synthesize fluorescent cyanine-3-labeled cRNA following the Agilent one-color Quick-Amp labeling protocol (Agilent Technologies, Amstelveen, The Netherlands). After this, the samples were hybridized on Agilent SurePrint G3 Human GE v2 8 × 60K Microarrays according to the manufacturer’s instructions. After washing, the microarrays were scanned using the Agilent G2505C DNA Microarray Scanner (Agilent Technologies). Raw data on pixel intensities was extracted from the scan images using Agilent’s Feature Extraction Software. The extracted data were checked for quality using ArrayQC (https://github.com/BiGCAT-UM/arrayQC_Module/; accessed on 20 December 2021), an in house developed quality control (QC) pipeline in R. For each spot, the following steps were taken: local background correction; flagging of bad spots, controls and spots with too low intensity; log2 transformation; and quantile normalization. Further preprocessing included the omission of flagged probes and merging of replicate probes based on median.

Overall, out of all performed microarrays, one of the Perfectha^®^ Deep samples did not pass the QC cut-off criteria and was, therefore, omitted from further analyses. Pre-processing of the data as described above was performed independently for the two sets of experiments:Phenion^®^ Full-Thickness Skin model exposed to RIVM preparations and Bio-Alcamid^®^;Phenion^®^ Full-Thickness Skin model exposed to Perfectha^®^ fillers.

#### 4.2.4. Gene Expression Analysis

Differentially expressed genes (DEGs) were selected to identify the modulated gene expression profiles caused by exposure of the skin models to the different dermal fillers. For each of the selected groups of the in vitro experiments, a paired analysis between the three biological replicates of the exposures and their controls was conducted using a moderated *t*-test of the Limma (Linear Models for Microarray Data) R-package. Cut-off criteria used with this test to define significance was a Benjamini-Hochberg adjusted *p*-value < 0.05. The adjusted *p*-value was used to account for multiple testing [38]. In addition, the number of genes with an absolute fold change (FC) > 1.5 (this corresponds with a log2 FC of 0.585) were captured.

#### 4.2.5. Pathway Analysis

The selected DEGs for each in vitro exposure were analyzed by an over-representation analysis (ORA) in PathVisio (https://www.pathvisio.org/; accessed on 20 December 2021). The statistical analysis in PathVisio provides a Z-score for each pathway. Pathways were considered significantly enriched if the Z-score was higher than 1.96, which is based on the assumptions that the data have a hypergeometric distribution; a Z-score of 1.96 agrees with a *p*-value of 0.05.

## Figures and Tables

**Figure 1 ijms-23-13046-f001:**
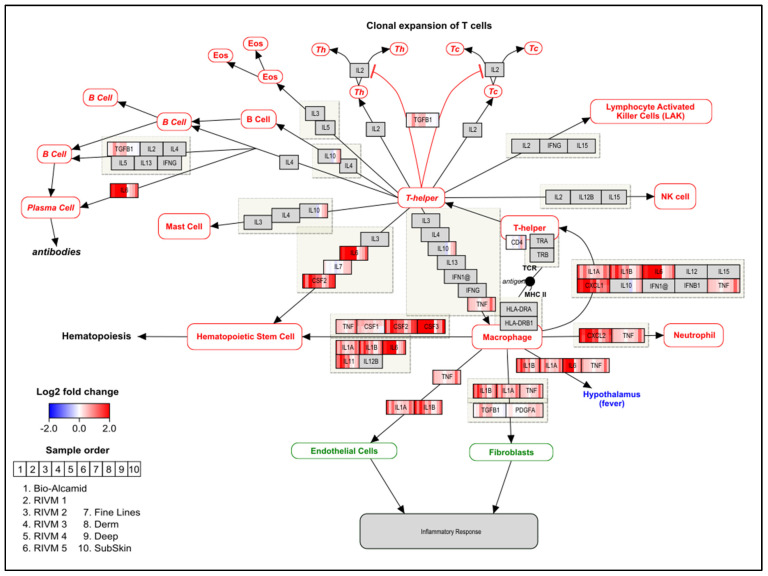
Visualization of gene expression changes for the Phenion^®^ Full-Thickness Skin model exposed to Bio-Alcamid^®^, RIVM preparations and Perfectha^®^ fillers on the “Cytokines and inflammatory response pathway”.

**Figure 2 ijms-23-13046-f002:**
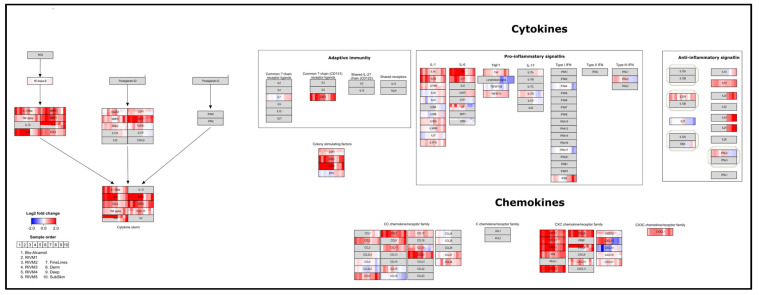
Visualization of gene expression changes for the Phenion^®^ Full-Thickness Skin model exposed to Bio-Alcamid^®^, RIVM preparations and Perfectha^®^ fillers on the “Overview of proinflammatory and profibrotic mediators”.

**Figure 3 ijms-23-13046-f003:**
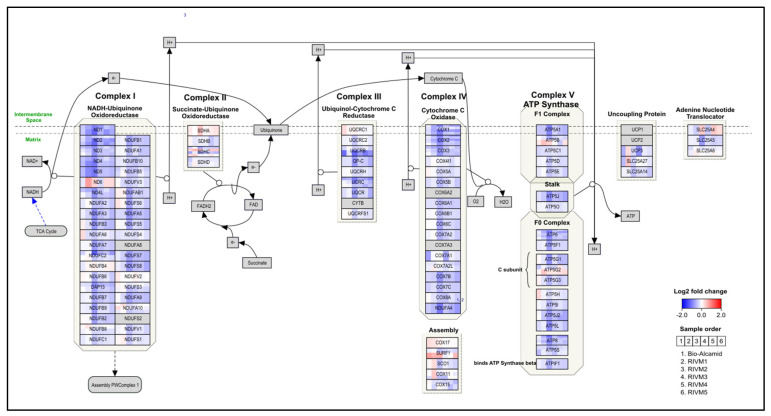
Visualization of gene expression changes for Phenion^®^ Full-Thickness Skin model exposed to Bio-Alcamid^®^ and RIVM preparations on the Electron transport chain: *OXPHOS system in mitochondria* pathway.

**Figure 4 ijms-23-13046-f004:**
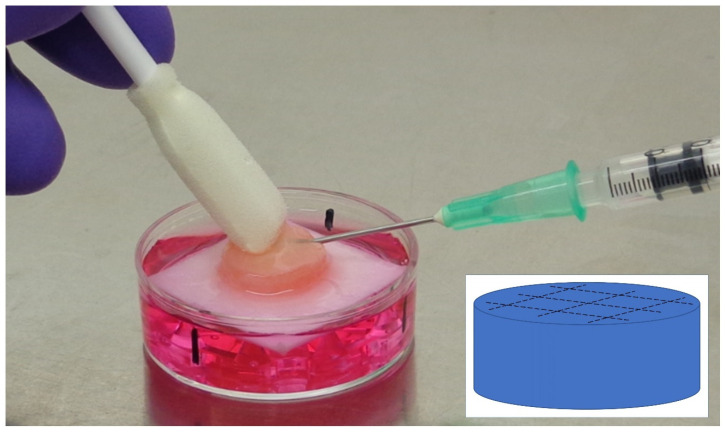
Injection of a dermal filler in the human Phenion^®^ Full-Thickness Skin Model. The inlay shows the used cross-hatching pattern.

**Table 1 ijms-23-13046-t001:** DEGs for the Phenion^®^ Full-Thickness Skin model exposed to Bio-Alcamid^®^ and RIVM ^1^ preparations. The number of genes significantly different compared to non-treated control cells is indicated in bold.

Phenion^®^	Bio-Alcamid^®^	RIVM 1 (1.9% CLG ^2^)	RIVM 2 (4.5% CLG)	RIVM 3 (7.6% CLG)	RIVM 4 (9.9% CLG)	RIVM 5 (14.4% CLG)
|FC| ≥ 1.5	1849	1374	2916	993	992	1375
Up-regulated	1260	1196	2236	493	789	1101
Down-regulated	589	178	680	500	203	274
**Adj. *p*-value < 0.05**	**666**	**117**	**4014**	**1**	**1**	**0**
|FC| and adj. *p*-value	490	98	2081	1	1	0
**Adj. *p*-value < 0.2**	**4379**	**2502**	**9862**	**275**	**16**	**1756**

^1^ RIVM = Rijksinstituut voor Volksgezondheid en Milieu (National Institute for Public Health and the Environment); ^2^ CLG = cross-linking grade (see Section 4 for further information).

**Table 2 ijms-23-13046-t002:** DEGs for the Phenion^®^ Full-Thickness Skin model exposed to Perfectha^®^ fillers. The number of genes significantly different compared to non-treated control cells is indicated in bold.

Phenion^®^	FineLines	Derm	Deep	SubSkin
|FC| ≥ 1.5	1889	1922	3173	3179
Up-regulated	809	919	1076	1056
Down-regulated	1080	1003	2097	2123
**Adj. *p*-value < 0.05**	**0**	**0**	**0**	**15**
|FC| and adj. *p*-value	0	0	0	15
**Adj. *p*-value < 0.2**	**8**	**0**	**1**	**1374**

**Table 3 ijms-23-13046-t003:** Top 10 significant pathways for the Phenion^®^ Full-Thickness Skin model exposed to Bio-Alcamid^®^. DEGs selected by moderated *t*-test with adjusted *p*-value < 0.05 were used for ORA in PathVisio. Pathways related to immune and inflammatory response are shown in bold.

Pathway	Z-Score	*p*-Value (Permuted) ^1^
**Cytokines and inflammatory response**	10.1	0.000
**Overview of proinflammatory and profibrotic mediators**	8.15	0.000
**COVID-19 adverse outcome pathway**	7.92	0.000
Zinc homeostasis	7.25	0.000
**Network map of SARS-CoV-2 signaling pathway**	6.67	0.000
**Prostaglandin signaling**	6.5	0.000
**SARS-CoV-2 innate immunity evasion and cell-specific immune response**	5.95	0.000
**Photodynamic therapy-induced NF-kB survival signaling**	5.53	0.000
**Development of pulmonary dendritic cells and macrophage subsets**	5.03	0.001
exRNA mechanism of action and biogenesis	5.01	0.000

^1^ *p*-values were rounded to 3 decimals.

**Table 4 ijms-23-13046-t004:** Top 10 significant pathways for the Phenion^®^ Full-Thickness Skin model exposed to RIVM 2 preparation. DEGs selected by moderated *t*-test with adjusted *p*-value < 0.05 were used for ORA in PathVisio. Pathways related to immune and inflammatory response and those related to cellular respiration are shown in bold and in italics, respectively.

Pathway	Z-Score	*p*-Value (Permuted) ^1^
*Electron transport chain: OXPHOS system in mitochondria*	7.69	0.000
*Oxidative phosphorylation*	7.56	0.000
Nonalcoholic fatty liver disease	5.99	0.000
**Overview of proinflammatory and profibrotic mediators**	4.37	0.000
*Mitochondrial complex I assembly model OXPHOS system*	4.36	0.000
**Prostaglandin signaling**	4.2	0.000
Glycolysis and gluconeogenesis	4.18	0.000
**IL-1 signaling pathway**	3.94	0.001
Bladder cancer	3.58	0.001
Cytokines and inflammatory response	3.58	0.000

^1^ *p*-values were rounded to 3 decimals.

**Table 5 ijms-23-13046-t005:** Top 5 significant pathways for the Phenion^®^ Full-Thickness Skin model exposed to RIVM 1, 3, and 5 HA preparations. DEGs selected by moderated *t*-test with adjusted *p*-value < 0.2 were used for ORA in PathVisio. Pathways related to immune and inflammatory response are shown in bold.

Pathway	Z-Score	*p*-Value (Permuted) ^1^
**RIVM 1 preparation**		
**miRNAs involvement in the immune response in sepsis**	6.67	0.000
Ganglio sphingolipid metabolism	5.99	0.000
**Prostaglandin signaling**	5.34	0.000
**Overview of proinflammatory and profibrotic mediators**	5.26	0.000
**SARS-CoV-2 innate immunity evasion and cell-specific immune response**	6.69	0.000
**RIVM 3 preparation**		
**Overview of proinflammatory and profibrotic mediators**	11.35	0.000
**Cytokines and inflammatory response**	10.76	0.000
**SARS-CoV-2 innate immunity evasion and cell-specific immune response**	10.02	0.000
**Photodynamic therapy-induced NF-kB survival signaling**	8.71	0.000
**Network map of SARS-CoV-2 signaling pathway**	8.32	0.000
**RIVM 5 preparation**		
**Photodynamic therapy-induced NF-kB survival signaling**	5.48	0.000
**Neuroinflammation**	5.45	0.000
Ethanol metabolism production of ROS by CYP2E1	4.83	0.000
Ketogenesis and ketolysis	4.06	0.000
**Photodynamic therapy-induced NFE2L2 (NRF2) survival signaling**	4	0.000

^1^ *p*-values were rounded to 3 decimals.

**Table 6 ijms-23-13046-t006:** Top 10 significant pathways for the Phenion^®^ Full-Thickness Skin model exposed to Perfectha^®^ SubSkin filler. DEGs selected by moderated *t*-test with adjusted *p*-value < 0.2 were used for ORA in PathVisio. Pathways related to immune and inflammatory response are shown in bold.

Pathway	Z-Score	*p*-Value (Permuted) ^1^
**Overview of proinflammatory and profibrotic mediators**	6.86	0.000
miRNA targets in ECM and membrane receptors	6	0.000
**Inflammatory response pathway**	5.36	0.000
Metabolism of alpha-linolenic acid	5.3	0.000
Omega-9 fatty acid synthesis	4.06	0.000
**Cytokines and inflammatory response**	3.97	0.000
miR-509-3p alteration of YAP1/ECM axis	3.77	0.003
Amino acid conjugation of benzoic acid	3.62	0.004
Butyrate-induced histone acetylation	3.62	0.004
**SARS-CoV-2 innate immunity evasion and cell-specific immune response**	3.47	0.002

^1^ *p*-values were rounded to 3 decimals.

**Table 7 ijms-23-13046-t007:** Product description of Perfectha^®^ Hyaluronic Acid dermal fillers.

	FineLines	Derm	Deep	SubSkin
Indication-for use	Fine lines & superfacial wrinkles	Moderate correction to face & lip contour	Deep wrinkles & furrows & lip augmentation	Volume augmentation cheeks, chin, jaw line
Injection site	Intradermal	Subcutaneous	Subcutaneous	Deep subcutaneous to supraperiosteal
Volume	0.5 mL	1 mL	1 mL	3 mL
No of particles (Gel/mL)	180,000	90,000	8000	2000
Duration of effect	4–6 months	6–12 months	8–12 months	12–18 months

**Table 8 ijms-23-13046-t008:** Overview of experimental synthesized cross-linked HA. Modification and cross-linking grade were determined by liquid chromatography–mass spectrometry (LC-MS).

Product	Amount HA (mg)	Amount BDDE (µL)	Volume (mL)	Modification Grade (%)	Cross-Linking Grade (%)
RIVM 1	496.6	60.0	25	10.7	1.9
RIVM 2	496.5	113.6	25	22.2	4.5
RIVM 3	500.7	170.5	25	31.5	7.6
RIVM 4	500.0	227.3	25	36.6	9.9
RIVM 5	500.5	454.5	25	42.9	14.4

## Data Availability

The data presented in this study are available in the article and the Supplementary Material. In addition, the raw and pre-processed microarray data have been uploaded to NCBI GEO (https://www.ncbi.nlm.nih.gov/geo/; accessed on 18 October 2022) and are available under accession numbers GSE216169 for the Phenion^®^ Full-Thickness Skin model exposed to RIVM preparations and Bio-Alcamid^®^ and GSE216167 for the Phenion^®^ Full-Thickness Skin model exposed to Perfectha^®^ fillers.

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
