# Peer review of "Transcriptomic Analysis in Human 3D Skin Model Injected with Resorbable Hyaluronic Acid Fillers Reveals Foreign Body Response"

_ijms, 2022, doi:10.3390/ijms232113046_

Round 1

Reviewer 1 Report

This manuscript is overall well written, and might bring new insights on the field. The methods are simple but well designed. 

I have only minor comments, as follows:

Some phrases in the Introduction are a little bit confusing. For instance, I didn´t understand the context of the phrase "patients can recover within weeks after treatment with hyaluronidase, an enzyme degrading HA-66 based fillers, resulting in a decrease of the swelling without leaving (severe) traces of the procedure". Therefore, the Introduction should be revised. Also, a resume of the results should be made.

In the discussion, the authors could also discuss the increase in the anti-inflammatory profile as the particle size incresead, with the augment of the phagocytic capacity of M2 macrophages (as well said by the authors "IL-10 is released by M2 macrophages"), maybe in an attempt to phagocyte foreign bodies.

The authors focused mostly on the inflammatory response, which is OK, but the readers, such as myself, would like to better understand why the SARS-CoV-2 and COVID-19 related pathways were studied. These results were not adressed in the discussion

Author Response

Dear reviewer,

We like to thank you for your constructive comments which we have addresses point-by-point below. 

Point 1.

Some phrases in the Introduction are a little bit confusing. For instance, I didn´t understand the context of the phrase "patients can recover within weeks after treatment with hyaluronidase, an enzyme degrading HA-66 based fillers, resulting in a decrease of the swelling without leaving (severe) traces of the procedure". Therefore, the Introduction should be revised. Also, a resume of the results should be made.

The indicated phrase is meant as an addition to the observation stated in the previous sentence about the patients being reluctant to provide biological samples (i.e. biopsies). Not only are the patients reluctant as taken biopsies could leave scaring, there is also no need to undergo surgery to remove the adverse effects caused by the HA-based fillers. No surgery and thus no opportunity to collect biopsies during the operational procedure. The HA-based fillers can be enzymatically removed, resulting in a decreased swelling without leaving any scars.

To make this more clear we adjusted the sentence: "Moreover, there is no need for an operational procedure at which biopsies could be taken, as treatment of the patients with hyaluronidase, an enzyme degrading HA-based fillers, results in a decrease of the swelling without leaving (severe) traces of the procedure [17-19]." (line 65-69)

Point 2.

In the discussion, the authors could also discuss the increase in the anti-inflammatory profile as the particle size incresead, with the augment of the phagocytic capacity of M2 macrophages (as well said by the authors "IL-10 is released by M2 macrophages"), maybe in an attempt to phagocyte foreign bodies.

Indeed the phagocytic capacity of M2 macrophages could increase with increased particle size. On the other hand M2 macrophages are also important for the formation of foreign body giant cells (FBGC). Possibly, the increased particle size could lead to the requirement of more macrophages to form the FBGC rather than direct phagocytosis of the particles. From the data, it is unclear what is exactly happening and further investigations would be needed. In addition, it should be noted that the 3D human skin model does not contain macrophages. Therefore, we would like to refrain from any speculation regarding the increase of IL-10 with increased particle size in relation with the phagocytic capacity of M2 macrophages.

To address the above points we included the following sentence: "Whether the increased particle size leads to more M2 macrophages, that are also involved in the formation of FBGC and thus more IL-10 expression, is unknown and remains to be determined. In the Phenion® Full-Thickness Skin model no macrophages are included and thus the origin of the changed IL-10 expression is unclear." (line 214-217)

Point 3.

The authors focused mostly on the inflammatory response, which is OK, but the readers, such as myself, would like to better understand why the SARS-CoV-2 and COVID-19 related pathways were studied. These results were not adressed in the discussion

The names of the SARS-CoV-2 and COVID-19 pathways are a bit misleading. These pathways contain many of the genes also found in the inflammatory and immune response related pathways, such as the "Overview of proinflammatory and profibrotic mediators" pathway shown in Figure 2, and therefore pop-up in the pathway analyses.

For example the "SARS-CoV-2 innate immunity evasion and cell-specific immune response" pathway (https://www.wikipathways.org/index.php/Pathway:WP5039) contains several members of the CC and CXC chemokine/receptor families. These genes are also present in the "Network map of SARS-CoV-2 signaling" pathway (https://www.wikipathways.org/index.php/Pathway:WP5115), which also contains many interleukins.

In Tables 3-6, pathways related to immune and inflammatory response are shown in bold. This includes the SARS-CoV-2 and COVID-19 pathways.

Since the SARS-CoV-2 and COVID-19 pathways represent inflammatory and immune response pathways they were not separately addressed in the discussion.

Reviewer 2 Report

Jennen et al. have conducted transcriptomic analysis in an in-vitro full-thickness skin model to find out if the injected HA fillers at different particle sizes and crosslinking levels can cause adverse effects. The hypothesis is partially confirmed, and the manuscript is well-written and well-organized. However, there are some concerns that need to be answered before it can be accepted.

1.     The authors should add a sentence or two in line 371 to explain why using Benjamini-Hochberg adjusted p-value, not the original p-value.

2.     Can authors explain if there are scientific reasons to use a less stringent P value = 0.2 as a cut-off, not 0.1 or 0.3?

3.     Line 136-138, 205-207, It is okay for authors to say higher expression for Subskin since there are at least 15 significant gene expression changes. But I did not see a significant difference in the Deep group based on the results in table 2, can the authors clarify?

4.     Are some p values in tables 3-5 really 0 or smaller than 0.0001? Please specify.

5.     Why is RIVM 2 at a 4.5% crosslinking level so different from the other RIVMs preparation? It is weird that RIVM 1 did not follow the trend. Have the authors conducted a biological repeat to confirm the experimental results in this study? And the authors should expand the discussion in line 249 to explain the underlying reasons or hypothesis for the cause of the highest expression in RIVM 2.

6.     The authors should put the crosslinking level next to RIVM when they were first introduced in the result section.

7.     Please clear the space between Lines 217, 239, and 255

8.     Authors should also discuss the limitations of using a 3D skin cell model compared to real human skin.

9.  Can authors re-upload supplemental materials as they can not be accessed?

Author Response

Dear reviewer,

We like to thank you for your constructive comments which we have addresses point-by-point below.

Point 1

The authors should add a sentence or two in line 371 to explain why using Benjamini-Hochberg adjusted p-value, not the original p-value.

The adjusted p-value is used to correct for multiple testing. We added the following sentence and added reference 38, the original Benjamini & Hochberg paper:

"The adjusted p-value was used to account for multiple testing [38]." (line 384)

Point 2

Can authors explain if there are scientific reasons to use a less stringent P value = 0.2 as a cut-off, not 0.1 or 0.3?

There is no scientific reason for selecting the less stringent adjusted p-value of <0.2. From our experience we know that a sufficient number of genes is needed to perform the pathway analyses. Below the number of genes using different adjusted p-values are shown. In bold are the number of DEGs indicated that would be sufficient for pathway analysis. We only used one less stringent adjusted p-value and not different ones for each separate experiment to keep the paper organized. As <0.1 was still too stringent in most cases and <0.3 too loose, we decided to use an adjusted p-value <0.2.

adj p-value

Bio-Alcamid®

RIVM 1

RIVM 2

RIVM 3

RIVM 4

RIVM 5

FineLines

Derm

Deep

SubSkin

0.05

666

117

4014

1

1

0

0

0

0

15

0.1

1936

584

6626

10

3

20

0

0

0

77

0.2

4379

2502

9862

275

16

1756

8

0

1

1374

0.3

6624

4902

12362

1064

272

4259

22

1

1

3545

Point 3

Line 136-138, 205-207, It is okay for authors to say higher expression for Subskin since there are at least 15 significant gene expression changes. But I did not see a significant difference in the Deep group based on the results in table 2, can the authors clarify?

There are two ways one can look at gene expression changes. On the one hand one can look at differentially expressed genes (DEGs) based on statistical significance and on the other hand one can look at the level of the changes, thus looking at the fold change. The number of DEGs are relevant for the pathway analysis per exposure, whereas the level of expression is interesting when comparing different exposures. In line 136-138 and 205-207 we looked at the latter, thus the level of the gene expression between exposures as shown in figure 1-3. Higher expression means in that sense higher for exposure A compared to exposure B. This is irrespective of whether the gene under investigation is a DEG or not for either exposure.

In the discussion (line 194-206) we have indicated the statistical significance where needed, else we mentioned the expression level differences between exposures.

Point 4

Are some p values in tables 3-5 really 0 or smaller than 0.0001? Please specify.

The PathVisio results only show rounded numbers with 3 decimals. A “0” (or “0.000” from the original output) is in fact a rounded number from a number smaller than 0.0005.

In tables 3-5 we have replaced the “0” by the original “0.000” and indicated that the p-values were rounded to 3 decimals.

Point 5

Why is RIVM 2 at a 4.5% crosslinking level so different from the other RIVMs preparation? It is weird that RIVM 1 did not follow the trend. Have the authors conducted a biological repeat to confirm the experimental results in this study? And the authors should expand the discussion in line 249 to explain the underlying reasons or hypothesis for the cause of the highest expression in RIVM 2.

This is a very good question. Based on the work by Keizers et al 2018 we expected a more profound adverse effect from the HA fillers with a higher modification and cross-linking grade, thus for RIVM 3-5, leading to more gene expression changes. If that would be the case RIVM 1 would follow the trend showing the least expression changes. However, RIVM 3-5 show hardly any gene expression changes. In the paper we already indicated a possible source for this, i.e. high viscosity and difficulties to inject these fillers. In line 256-257 we included a reference to the study by Keizers et al. 2018, i.e. “as was expected based on the study by Keizers et al. [34]”.

In addition, for each HA filler 3 independent experiments were performed, thus for each exposure data 3 biological replicates were available. In line 333-335 this information was already included “Exposures lasted for 24 hours at 37oC and were performed in triplicate. Each set of exposures was performed on a different date using a fresh batch of tissue models, all constructed from the same donor.” We added the following sentence to this information to emphasize that we have 3 biological replicates: “Thus, for each exposure three biological replicates were made.” (line 336). Furthermore, we now also mention the 3 biological replicates in the sub-section on the gene expression analysis (line 381)

Point 6

The authors should put the crosslinking level next to RIVM when they were first introduced in the result section.

The RIVM fillers were first mentioned in Table 1. Therefore, we included the cross-linking grade (CLG) in this table, with a reference to Table 8 in the Material and Methods section.

Point 7

Please clear the space between Lines 217, 239, and 255

The space between the indicated lines has been removed.

Point 8

Authors should also discuss the limitations of using a 3D skin cell model compared to real human skin.

You are quite right that the currently used 3D skin cell model has some limitations compared to real human skin. In our model no macrophages are incorporated and it is a static model. However, a major advantage of a full-thickness skin model is that the availability and quality of the model can be more standardized than for human skin samples.

To overcome the short comings of the 3D skin model compared to real human skin, we have indicated under the conclusion how the current 3D skin model can be improved to better resemble the real human skin:

“In addition, our current setup of the experiments with the human Phenion® Full-Thickness Skin model can be further improved by including macrophages in the medium [35] and/or by introducing vascularization to the system by which a more physiological transport of nutrients can be established equivalent to real human skin tissue [36].” (line 276-280)

Point 9

Can authors re-upload supplemental materials as they can not be accessed?

The zipped file containing the supplementary tables turned out to be corrupted and has now been replaced.

Round 2

Reviewer 2 Report

I have no further concerns about the manuscript